# Impact of productive social safety net on households' vulnerability to poverty in Tanzania

Basil Msuha [1,2]*, Luitfred D. Kissoly[2]

1 Head Office (Makutupora), Tanzania Agricultural Research Institute (TARI), Dodoma, Tanzania,
2 Department of Economics and Social Studies, School of Spatial Planning and Social Sciences, Ardhi University, Dodoma, Tanzania

* basil.msuha@gmail.com

## Abstract

Social safety nets are expanding in Sub-Saharan Africa. While proponents perceive them as a means to combat poverty and vulnerability, opponents view them as wasteful use of scarce public resources and do not significantly overcome poverty. Previous studies have primarily focused on assessing the impact of these policies on current poverty levels, with insufficient evidence available regarding their impact on future poverty, which deserves equal attention. We drew on the Tanzanian 2017–18 Household Budget Survey, comprising 9,463 households to evaluate the impact of productive social safety net (PSSN) program on households' vulnerability to poverty (VP). The VP was evaluated using vulnerability as expected poverty (VEP), whereas the impact was estimated using Instrumental Variable (IV) method. We found that PSSN reduces household VP by 13.4%, suggesting that it is an effective policy instrument for reducing poverty and vulnerability. Notably, the estimated impacts were greater for households enrolled in conditional cash transfer (CCT) and public work (PW) combined, suggesting that a package of CCT and PW is likely to have a more substantial impact within the realm of social safety nets. Our findings offer evidence in favor of policies that promote the broader expansion of social safety nets as anti-poverty policy instruments.

## 1. Introduction

Poverty is a global quandary, and despite efforts to end it, progress has been slow. In some parts of the world, the quandary has worsened. Notably, in sub-Saharan Africa (SSA), poverty rates have remained stagnant, while the number of individuals experiencing extreme poverty continues to grow. For instance, in 1990, the region accounted for 14% of the global poor population, and by 2019, this figure had escalated to 57% [1]. According to recent poverty projections, the continent is expected to constitute the largest share of the world's poor by 2030 [2–4].

To combat poverty, each government in sub-Saharan Africa *inter alia*, is implementing at least one type of social safety net [5, 6]. Despite the widespread implementation of social safety nets, there are two opposing viewpoints regarding their effectiveness [7, 8]. Proponents of social

**Data Availability Statement:** All relevant data are within the paper and its Supporting information files.

**Funding:** This work was supported by the Economic and Social Research Foundation (ESRF)

under the Small Grants Initiative for Capacity Development in Impact Evaluation Research. This initiative is funded by the William and Flora Hewlett Foundation, and falls under the auspices of the Impact Evaluation Laboratory within the ESRF.

**Competing interests:** The authors have declared that no competing interests exist.

safety nets perceive them as a means to combat poverty and vulnerability [9, 10]. They are viewed as effective anti-poverty policy instruments [9, 10] that can substantially support the concept of graduation by assisting extremely poor individuals or households in escaping poverty and the risk of falling or remaining poor in the future [11–13]. Opponents view social safety nets as short-term palliatives and wasteful use of scarce public resources [14], subsequently, do not significantly overcome poverty [15–17]. Instead, they foster dependency [18, 19].

At the center of the debate, there has been a notable increase in empirical studies globally [20] and in sub-Saharan Africa [9] that have evaluated the effectiveness of social safety nets. The majority of existing studies have focused on assessing the impacts on current poverty—*ex-post evidence* [9, 20, 21] leaving insufficient evidence regarding their impact on future poverty—*ex-ante evidence* [22]. Both ex-post and ex-ante evidence are important for gauging the effectiveness of anti-poverty interventions [23] because poverty is dynamic and can change over time. Individuals or households may be non-poor in the current year, but may be poor in the following year and vice versa [12]. Consequently, effective anti-poverty interventions should address both aspects, as they are two sides of the same coin [24]. In addition, previous research on the impact of social safety nets has mostly focused on conditional cash transfers (CCTs), with less emphasis on public work (PW) schemes [25], thus limiting our broader understanding of their effectiveness.

This study adds to the existing body of literature in two ways. First, it extends previous studies on the impact of social safety nets to include vulnerability to poverty. By doing so, it aims to construct new knowledge, much like adding a new brick to a wall. This knowledge is crucial for policymakers as it provides complete insights into the current and future poverty impact of social safety nets—the two sides of the same coin. Second, this study explores the impact of public works and the implications of combining conditional cash transfer (CCT) and public work (PW) within social safety nets. This knowledge is essential for designing effective social safety net programs.

The remainder of this paper is organized as follows. Section two explores the fundamental premise of social safety nets and delves into the conceptual framework that connects them to vulnerability to poverty. This section also explores empirical studies on this topic and identifies the existing research gaps. Section three expounds on the data sources and the estimation strategy. Section four presents the results, and section five offers the discussion. Section six details the limitations of the study and potential areas for further research, while section seven provides the conclusions and policy implications.

## 2. Social safety nets and Vulnerability to Poverty (VP)

To tackle the persistent problem of poverty, numerous countries in sub-Saharan Africa have implemented at least one form of social safety net, including conditional cash transfers (CCTs), public works (PWs), or a combination of both [5, 6]. For example, since 2000, Tanzania has implemented more than 12 types of social safety nets comprising cash-based transfers, in-kind transfers, and public work [26]. The Tanzania's Productive Social Safety Nets (PSSN) is one of the most extensive programs in Africa, comprised of conditional cash transfer (CCT) and public works (PWs), enrolling more than 1.1 million poor households. The CCT offers a transfer value of up to 16.3 USD per month per household, while the PWs scheme provides a daily rate of 1.35 USD for a maximum of 15 days of paid work per month per household, limited to four months during the annual lean season [10]. The Tanzanian Social Action Fund (TASAF) oversees the implementation of the PSSN program.

Globally, social safety nets are considered to be effective policy instruments for combating poverty and vulnerability [10, 13, 27]. It is suggested that enrolling extremely poor and

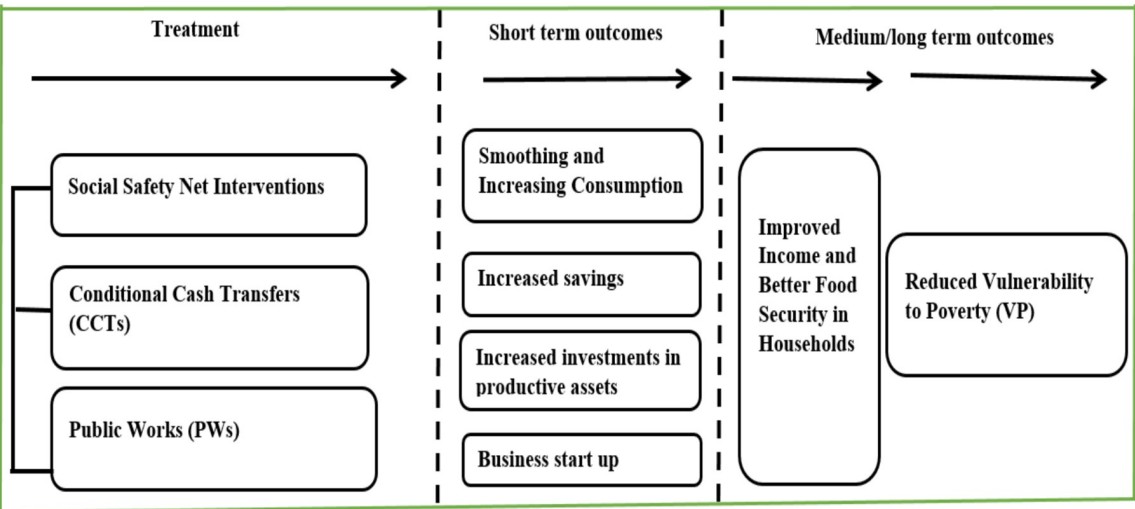

**Fig 1. Conceptual framework for evaluating impact of social safety nets on vulnerability to poverty.**

vulnerable households in social safety net programs leads to the provision of cash transfers, which have immediate effects on increasing and smoothing consumption [28, 29], promoting savings, starting businesses, and investing in productive assets, [30] as illustrated in Fig 1. These short-term outcomes result in improved household income and consumption [31], ultimately leading to a reduction in poverty and vulnerability [13]

Although social safety nets hold the premise of poverty and vulnerability reduction, there is ongoing disagreement regarding their effectiveness [7, 8, 14, 32, 33]. To the best of our knowledge, most empirical studies examining the impact of social safety net programs have indicated the impact by linking conditional cash transfers to current poverty levels [20] without considering future poverty. For example, in an extensive review, Bastagli et al. [21] analyzed the impacts of 37 social safety net programs on poverty in sub-Saharan Africa. Their findings offer strong empirical support for the significant impact of social safety nets on poverty reduction. The World Bank [34] shows that the productive social safety net (PSSN) program has been successful in reducing poverty levels by 6.9 percentage points among Tanzanian participants.

Research trends beyond sub-Saharan Africa also demonstrate a consistent pattern. According to Perova and Vakis [35], Juntos participants in Peru experienced a 14 percent reduction in poverty, whereas Cheema et al. [36] reported a 22 percent reduction in poverty in the Benazir Income Support Program (BISP) in Pakistan. Bastagli et al. [21] reviewed 36 social safety net programs in Latin America, 18 in South Asia, 10 in East Asia and the Pacific, three in Europe and Central Asia, five in North America, and four in the Middle East and North Africa, demonstrating a significant impact on poverty reduction among beneficiaries. Generally, the existing body of research has not provided significant insights into the relationship between social safety nets and vulnerability to poverty. Consequently, our understanding of the effectiveness of these programs remains limited.

## 3. Materials and methods

### 3.1 Study setting and design

This study estimates the impact of productive social safety nets (PSSN) on household vulnerability to poverty (VP) in mainland Tanzania. Tanzania comprises 31 administrative regions

and 195 local government authorities (LGAs). Of these, 184 are located on the mainland in 26 regions, and 11 LGAs in five regions are in Zanzibar. The PSSN program was implemented across all LGAs, enrolling 1.118 million households, with over one million situated on the mainland. This study focuses exclusively on Tanzania Mainland, where the household budget survey (HBS) includes a specific module for collecting data to monitor the progress of the productive social safety net (PSSN) program.

While the PSSN program design was based on randomized controlled trials (RCT), HBS data, generation did not follow the RCT protocol. Consequently, a quasi-experimental design was chosen as the next best option to address the lack of a random assignment. This approach helps identify a comparison group closely resembling the PSSN participants to estimate what the VP would have been without PSSN intervention (the counterfactual).

## 3.2 Data source

To estimate the impact of productive social safety nets (PSSN) on household vulnerability to poverty (VP) in mainland Tanzania, we drew on the 2017–18 Household Budget Survey (HBS) dataset. This dataset encompasses all 26 regions and 184 local government authorities in mainland Tanzania, providing a comprehensive and nationally representative picture. The HBS dataset is widely acknowledged as a preeminent source of data for evaluating antipoverty initiatives and measuring poverty levels [37]. Therefore, it was deemed the most suitable dataset for our study.

The 2017–18 HBS questionnaire includes a specific module on PSSN with a filter question (22E.01) that inquired whether a household received cash transfers from the TASAF, excluding wages from public works. Another filter question (22E.06) asked if a household received cash transfers from the TASAF to participate in the public works program. Information from these questions was used to generate the treatment variables. The HBS employed a two-stage cluster-sampling design. Primary sampling units (PSUs) or enumeration areas were chosen from the 2012 population and housing census frame (2012 PHC). A total of 796 PSUs were selected, and the households within these PSUs were listed before sampling. A total of 51 sampling strata were established and a representative probability sample of 9,552 households was chosen. Of the 9,552 households selected, 9,463 were successfully interviewed.

## 3.3 Estimation strategy

In a quasi-experimental design, it is possible to estimate the impact of programs by using matching methods. These methods encompass a range of statistical techniques, referred to as matching estimators, or matching methods that employ various matching algorithms [38]. These include propensity score matching (PSM), regression adjustment (RA), inverse probability weights (IPW), doubly robust methods (inverse-probability-weighted regression adjustment-IPWRA and augmented inverse-probability weights (AIPW) [39], and kernel-based propensity score matching (kmatch) [40].

Matching methods employ statistical techniques to create "an artificial comparison group" from non-treatment observations that is most similar to the treated group [38]. These methods rely on participants' observed characteristics, assuming "no unobserved differences" that affect outcomes between groups. Consequently, matching methods are typically combined with other approaches that address unobservables, such as instrumental variables (IV) and difference-in-differences (DID) [38]. The DID is suitable when we have data for both the treated and control groups before and after treatment, whereas the IV is suitable for evaluating the impact of programs when not everyone complies with the assignment rule (imperfect compliance) [38, 40].

In Tanzania, households with monthly food consumption per adult equivalent below TZS 33,748 are deemed extremely poor and most vulnerable, qualifying them for the PSSN program [41]. Over four million Tanzanians meet the eligibility criteria for PSSN owing to extreme poverty and vulnerability [37]. Nevertheless, only approximately 1.1 million households are enrolled in the program [10]. This indicates non-compliance with the assignment rule, which in turn suggests that the use of instrumental variable (IV) techniques to assess the program's impact is viable.

To avoid the identification problem, we employed the optimal instrumental variable strategy recommended by Wooldridge [42] and Xu [43] rather than adopting a direct approach [44]. We conducted a Probit regression analysis in which we used the binary treatment variable (PSSN) as the dependent variable, and the eligibility rule (Z) and covariates (Xi) of interest as independent variables. The results of the analysis are presented in S1 Appendix. Subsequently, we employed the predicted probability of PSSN as the instrument for the treatment model using the two-stage least squares (2SLS) method. The probability of participation $[P(PSSN_i|Z_i, X_i)]$ serves as its orthogonal projection, and is considered to be the optimal instrumental variable [42]. Specifically, we applied the two-stage least squares approach to the IV framework, as follows:

Stage I

$$PSSN_i = \delta_0 + \delta_1 Z_i + \delta_2 X_i + \varepsilon_i \tag{1}$$

Stage II

$$VP_i = \alpha_0 + \alpha_1 \char`\^ PSSN_i + \alpha_2 X_i + \varepsilon_i \tag{2}$$

Where:

$VP_i$ = Denotes vulnerability to poverty (outcome variable) estimated using the vulnerability as expected poverty (VEP) approach suggested by Chaudhuri et al. [45], Gaiha and Imai [46], and Günther and Harttgen [47]. A more detailed estimation procedure can be found in Msuha and Kissoly [48].

$PSSN_i$ = Dummy for treatment; equals to 1 = if household $i$ is enrolled in PSSN program and 0 otherwise

$Z_i$ = Denotes eligibility rule = 1 if household $i$ is eligible to PSSN program

$\char`\^ PSSN_i$ = Probability of being enrolled in the PSSN program for household $i$ which is the estimated instrumental variable (IV) generated in Eq (1)

$X_i$ = Denotes the set of covariates

$\varepsilon_i$ = Part of VP that is not explained by PSSN participation or by observed characteristics

## 4. Results

### 4.1 Descriptive results

The descriptive statistics are presented in Tables 1 and 2. The findings in Table 1 show that the majority of households (approximately 57.7 percent) were headed by men, while 42.3 percent were headed by women. A comparison of the characteristics between male-headed households (MHHs) and female-headed households (FHHs) among PSSN households reveals notable differences. For example, dependency ratio in FHHs is notably higher than that in MHHs, at 1.67 versus 1.55, respectively. Furthermore, the average family size in MHHs was significantly

**Table 1. Descriptive results.**

| Variable | Pooled | PSSN | Non-PSSN | PSSN | | t statistic/ | PSSN | | t statistic/ |
|---|---|---|---|---|---|---|---|---|---|
| | | | | Female | Male | chi2 | Urban | Rural | chi2 |
| Age of HH head (years) | 47.366 | 55.169 | 46.557 | 58.33 | 52.85 | -53.8(0.00) | 56.36 | 54.83 | -13.5(0.00) |
| Dependency ratio (ratio) | 1.548 | 1.599 | 1.543 | 1.67 | 1.55 | -16.0(0.00) | 1.34 | 1.67 | 37.4(0.00) |
| Family size (adult equivalent) | 4.965 | 5.174 | 4.944 | 4.57 | 5.62 | 75.5(0.00) | 5.02 | 5.22 | 9.0(0.00) |
| Sex (1 = Male) † | 0.760 | 0.577 | 0.779 | | | | 0.44 | 0.61 | 186.5(0.00) |
| Location (1 = rural) † | 0.682 | 0.776 | 0.672 | 0.71 | 0.83 | 186.5(0.00) | | | |
| fps (1 = extremely poor) † | 0.080 | 0.123 | 0.076 | 0.10 | 0.14 | 36.8(0.00) | 0.12 | 0.12 | 0.02(0.727) |
| ps_low (1 = poor) † | 0.264 | 0.380 | 0.252 | 0.33 | 0.42 | 66.6(0.00) | 0.40 | 0.37 | 4.3(0.00) |
| Employment (employee/self) † | 0.746 | 0.683 | 0.753 | 0.69 | 0.68 | 1.9(0.00) | 0.77 | 0.66 | 88.7(0.00) |
| Remittances (1 = received) † | 0.109 | 0.139 | 0.106 | 0.17 | 0.12 | 40.3(0.00) | 0.10 | 0.15 | 25.3(0.00) |
| Business (1 = own business) † | 0.222 | 0.188 | 0.226 | 0.19 | 0.19 | 0.01(0.7638) | 0.23 | 0.18 | 24.2(0.00) |

† Variables involved in chi2 test statistic; Figures in parentheses are p-values.

larger (5.6 adult equivalent) than that in FHHs (4.6 adult equivalent). The findings indicate that, on average, FHHs were 58.3 years old, while MHHs were, on average, 52.9 years old.

The distribution of the study population varied substantially between FHHs and MHHs, as well as between rural and urban locations. Notably, a higher dependency ratio was observed in the rural areas (1.67) than in the urban areas (1.34). Additionally, households with larger family sizes were more prevalent in rural areas (5.22) than in urban areas (5.02). The findings further show that 26.4 percent of the population lived below the basic poverty line and 8 percent lived below the food poverty line. A substantial proportion of PSSN households were located in rural areas (77.6 percent) compared to urban areas. This result aligns with the findings of the National Bureau of Statistics [37] concerning the distribution of poor populations on Tanzania's mainland, where the majority of the poor live in rural areas.

The descriptive results in Table 2 reveal a noteworthy pattern: vulnerability to poverty (VP) is significantly lower among FHH than their MHH counterparts (P< 0.001). Similarly, the prevalence of vulnerability was significantly higher in rural areas than in urban areas (P< 0.001). The vulnerability is significantly lower among households enrolled in CCT + PW combined compared to their counterparts enrolled in CCT only.

## 4.2 Econometric results

Table 3 presents the IV estimation results for the impact of PSSN on household VP across Panels A, B, and C. S2 Appendix presents the complete IV regression results. Panel A displays the impact results for households enrolled in the PSSN program and received either CCT only or

**Table 2. Vulnerability status by sex and location.**

| VP status | Sex | | | Location | | |
|---|---|---|---|---|---|---|
| | Female | Male | t statistic | Rural | Urban | t statistic |
| Panel A: All households | 0.245 | 0.271 | 63.1(0.000) | 0.322 | 0.141 | 505(0.000) |
| Panel B: PSSN households | 0.266 | 0.318 | 53.2(0.000) | 0.328 | 0.186 | 123.4(0.000) |
| Panel C: CCTs only households | 0.267 | 0.322 | 51.99(0.000) | 0.332 | 0.187 | 123.4(0.000) |
| Panel D: CCTs + PWs households | 0.243 | 0.304 | 46.0(0.000) | 0.299 | 0.202 | 46.2(0.000) |

Except for the t-statistics and parentheses that contain p-values, all other figures represent vulnerability indices.

**Table 3. Impact of PSSN on Vulnerability to Poverty (VP).**

| Outcome: VP | Instrumental Variable (IV) |
|---|---|
| | (ATT) |
| **Panel A: Overall impact** | |
| PSSN | -0.134*** |
| | (0.032) |
| **Panel B: Conditional cash transfers** | |
| CCT Only | -0.140*** |
| | (0.034) |
| **Panel C: Conditional cash transfers and public works** | |
| CCT+PW | -0.389*** |
| | (0.121) |

Note: ATT refers to average treatment effect on the treated. The figures in parentheses represent standard errors, and the following hold true for statistical significance:

***$P < 0.01$,

**$P < 0.05$, and

*$P < 0.1$.

The analysis is adjusted for age, age squared, sex, adult equivalent, adult equivalent squared, dependence ratio, marital status, location, livestock, employment, remittances, income sources, food assistance, health subsidies, and business ownership.

The complete IV regression results are presented in S2 Appendix.

In Panel A, PSSN is an indicator that equals one if the household was enrolled in the PSSN program and received either CCT only or CCT + PWs. In Panel B, CCT equals one if the household received CCT only, and CCT + PW is equal to one if the household received CCT and an additional transfer in the form of PW.

CCT+PW, and Panel B presents the impact results for households that received only CCT and CCT+PW for those who received CCT and an additional transfer in the form of PW.

The IV technique requires the instrument to be correlated with the endogenous explanatory variable (treatment) but uncorrelated with the error term (affect the outcome only through the treatment). To ensure the accuracy of our findings, we performed various post-estimation diagnostic tests.

We initially performed a Durbin-Wu-Hausman (DWH) specification test [49] to determine whether the treatment variable was an endogenous component of the model. The test results rejected the null hypothesis of exogeneity, providing evidence that the treatment variable was an endogenous element within the model. The first-stage regression statistic test for all three estimations was significant at the 1 percent level, and the first-stage $F$-statistic value exceeded 10 [50]. Consequently, the null hypothesis of weak instruments was rejected, suggesting that IV was strongly correlated with the treatment variable at the 1 percent level of significance. Sargan's [51] and Basmann's [52] χ2 tests of overidentification restrictions indicated that the instrument was uncorrelated with the error term; thus, the IV was valid and the model was correctly specified. S3 Appendix presents the detailed test findings.

The findings in Table 3 show that enrollment in the PSSN program led to a statistically significant reduction in vulnerability to poverty among extremely poor households (ATT = -0.134, P < 0.001). This suggest that participation in the PSSN program decreased the likelihood of falling or remaining in poverty by 13.4 percentage points. Moreover, the decrease was 14 percent for participation in CCT only (P < 0.001) and 38.9 percent for participation in CCT+PW (P < 0.001). It is important to note that the impact was found to be significantly

higher among households that participated in both CCT and PW programs (ATT = -0.389, P < 0.001), as shown in Table 3, Panel C.

These results imply that, within the social safety net space, policy efforts to support chronically poor households from the risk of falling or remaining in poverty in the future may have a significantly greater impact, especially when CCTs and PWs schemes are combined. Such efforts may have less impact when the two are separated.

## 5. Discussions

This study found evidence consistent with the positive and significant impact of social safety nets on households' vulnerability to poverty. This holds true for both conditional cash transfer (CCT) and conditional cash transfer combined with public work (CCT + PW). Specifically, the results indicate that when an extremely poor household is enrolled in a productive social safety net (PSSN) program, regardless of whether it is CCT or CCT + PW combined, the probability of falling or remaining in poverty in the future decreases by 13.4 percentage points. The results align with those of proponents of social safety nets, who argue that they are effective policy instruments in combating poverty and vulnerability among poor and vulnerable households [9, 10].

Our findings align with the claim that social safety nets are effective in reducing vulnerability [7, 8] and play a critical role in promoting graduation by offering support to the extremely poor in escaping poverty and the risk of falling or remaining poor in the future [11, 12, 53]. The results corroborate the main objective of social safety nets, which is to alleviate both current and future poverty, as they are two sides of the same coin [24].

Notably, the combined impact of CCT and PW (CCT + PW) has a more significant influence on reducing household vulnerability to poverty compared to CCT alone. This might be attributed to the fact that in addition to CCT, PWs possess three potential impact channels: wage transfers, asset creation, and skills development [54, 55]. Consequently, the joint vulnerability reduction impact is likely to surpass that of CCT alone. Although it would be desirable to analyze the separate impacts of CCT only, PW only, and the combination of both, this was not possible owing to data limitations. Nevertheless, this finding offers valuable insights into the benefits of PWs and combining CCTs and PWs within social safety networks. Similarly, the findings are consistent with the assertions made by PW proponents, who posit that they are effective in addressing vulnerability in both low- and middle-income countries [25, 28, 56].

Collectively, the results of our study are consistent with the notion that social safety nets are useful policy tools for reducing vulnerability [7, 8] and align with Sustainable Development Goal 1.3, which advocates the establishment of effective social protection systems to aid poor and vulnerable groups [57]. Thus, our findings contradict the notion that social safety nets are short-term palliatives, wasteful use of limited public resources [14], and ineffective in alleviating poverty [15–17]. Our results indicate that social safety nets are not only effective in reducing poverty levels, as has been found in previous research conducted elsewhere [9, 20, 21] and in Tanzania [58, 59], but they also reduce vulnerability to poverty.

## 6. Limitations and suggestions for future research

This study sought to estimate the relative impacts of conditional cash transfer (CCT) and public work (PW) on households' vulnerability to poverty. Consequently, our analysis would be more complete with an understanding of the separate impacts of CCT only, PW only, and CCT + PW. This is not the case; instead, we have shown the relative impact of CCT and CCT + PW combined because of the data limitations. Currently, the PSSN program does not implement a public works (PW) scheme as a standalone treatment arm, which is why the

2017–18 HBS were unable to gather such data. More research, particularly randomized controlled trials (RTCs) with PW as a distinct treatment arm, is essential for thorough impact evaluation. More than 91 countries worldwide have implemented public work programs [25, 60, 61], underscoring the need for a more in-depth examination of their effectiveness.

## 7. Conclusions and policy implications

Social safety nets are expanding in African countries as policy instruments for combating poverty and vulnerability. While the impact of these instruments on vulnerability to poverty (VP) remains far from clear, existing studies have mainly focused on assessing the impacts on current poverty—*ex-post evidence*–leaving insufficient evidence regarding their impact on future poverty—*ex-ante evidence*. This study draws on the Tanzanian 2017–18 Household Budget Survey dataset, comprising 9,463 households to estimate the impact of productive social safety nets (PSSN) program on households' vulnerability to poverty (VP). Specifically, this study explores the impact of conditional cash transfer (CCT) and the implications of combining CCT and Public Works (PW) within a social safety net space. This knowledge is essential for designing effective social safety net programs. Our analysis yielded results indicating that conditional cash transfer (CCT) and public work (PW) programs have a significant impact on reducing households' vulnerability to poverty (VP). Notably, the combined impact of CCT and PW was greater than that of CCT alone.

Based on these conclusions, we can infer several policy implications. *First*, findings imply that social safety nets are not only effective in reducing poverty levels, as has been established in prior research, but they can also reduce the likelihood of falling into or remaining in extreme poverty among poor and vulnerable households in the future. It is myopic to view these programs as merely short-term palliatives that waste scarce public resources and do little to alleviate poverty. *Second*, combining conditional cash transfer (CCT) and public work (PW) programs within the realm of social safety nets is likely to have a more significant impact on reducing vulnerability to poverty compared to CCT programs alone.

## Supporting information

**S1 Appendix. Probit model for obtaining predicted probability of PSSN.**
(DOCX)

**S2 Appendix. Complete IV regression results on the impact of PSSN on households' vulnerability to poverty.**
(DOCX)

**S3 Appendix. Postestimation: First-stage F-statistics and Durbin-Wu-Hausman (DWH) specification tests.**
(DOCX)

**S1 File.**
(DO)

**S2 File.**
(DTA)

## Author Contributions

**Conceptualization:** Basil Msuha.

**Formal analysis:** Basil Msuha.

**Methodology:** Basil Msuha.

**Supervision:** Luitfred D. Kissoly.

**Validation:** Luitfred D. Kissoly.

**Writing – original draft:** Basil Msuha.

**Writing – review & editing:** Luitfred D. Kissoly.

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
