## [Decision Letter · Decision Letter 0]

2 Jul 2024

PONE-D-24-02693Impact of Productive Social Safety Nets on Households’ Vulnerability to Poverty in TanzaniaPLOS ONE

Dear Dr. Msuha,

Thank you for submitting your manuscript to PLOS ONE. After careful consideration, we feel that it has merit but does not fully meet PLOS ONE’s publication criteria as it currently stands. Therefore, we invite you to submit a revised version of the manuscript that addresses the points raised during the review process.

We look forward to receiving your revised manuscript.

Kind regards,

Muhammad Khalid Bashir, PhD

Academic Editor

PLOS ONE

2. Please ensure that you refer to Figures 1 and 2 in your text as, if accepted, production will need this reference to link the reader to the figure.

Reviewers' comments:

Reviewer's Responses to Questions

**Comments to the Author**

1. Is the manuscript technically sound, and do the data support the conclusions?

Reviewer #1: Yes

Reviewer #2: Yes

2. Has the statistical analysis been performed appropriately and rigorously? 

Reviewer #1: Yes

Reviewer #2: Yes

3. Have the authors made all data underlying the findings in their manuscript fully available?

Reviewer #1: Yes

Reviewer #2: Yes

4. Is the manuscript presented in an intelligible fashion and written in standard English?

Reviewer #1: Yes

Reviewer #2: Yes

5. Review Comments to the Author

Reviewer #1: This study empirically investigates the effect of SSNs on household’s vulnerability to poverty in Tanzania

To further improve the quality of the manuscript, the following suggestions are proposed:

1. The reviewed literature does not critically analyze the existing literature on (vulnerability to) poverty or on SSNs-Poverty nexus

2. The study does not provide suggestions for future research

3. The figures, given in the end, are blurred (not clear)

Reviewer #2: the study on Impact of Productive Social Safety Net on Households’ Vulnerability to Poverty in

Tanzania is very informative and healthy exercise for the readers on the poverty reduction topic particularly.

The paper is scientifically sound and organized in proper manners.

I have some suggestions for the authors:

1) The introduction part is more concise please elaborate it little bit more and clear the research gap of the study.

2) The literature review part of the paper should also be enhanced. It is too small.

3) The discussion part need more citations and references as well and discuss each variable impact on poverty vulnerability.

4) provide the suggestion on the basis of the findings that authors found in the study not to be generalized but specific policy implications.

6. PLOS authors have the option to publish the peer review history of their article (what does this mean?). If published, this will include your full peer review and any attached files.

Reviewer #1: No

Reviewer #2: No

---

## [Author Response · Author response to Decision Letter 0]

25 Jul 2024

Dear Muhammad Khalid Bashir, PhD

Our response to reviewer is submitted in the submission system

---

## [Decision Letter · Decision Letter 1]

30 Jul 2024

Impact of Productive Social Safety Nets on Households’ Vulnerability to Poverty in Tanzania

PONE-D-24-02693R1

Dear Dr. Msuha,

We’re pleased to inform you that your manuscript has been judged scientifically suitable for publication and will be formally accepted for publication once it meets all outstanding technical requirements.

Kind regards,

Muhammad Khalid Bashir, PhD

Academic Editor

PLOS ONE

Additional Editor Comments (optional):

Reviewers' comments:

Reviewer's Responses to Questions

**Comments to the Author**

1. If the authors have adequately addressed your comments raised in a previous round of review and you feel that this manuscript is now acceptable for publication, you may indicate that here to bypass the “Comments to the Author” section, enter your conflict of interest statement in the “Confidential to Editor” section, and submit your "Accept" recommendation.

Reviewer #1: All comments have been addressed

Reviewer #2: All comments have been addressed

2. Is the manuscript technically sound, and do the data support the conclusions?

Reviewer #1: Yes

Reviewer #2: Yes

3. Has the statistical analysis been performed appropriately and rigorously? 

Reviewer #1: Yes

Reviewer #2: Yes

4. Have the authors made all data underlying the findings in their manuscript fully available?

Reviewer #1: Yes

Reviewer #2: Yes

5. Is the manuscript presented in an intelligible fashion and written in standard English?

Reviewer #1: Yes

Reviewer #2: Yes

6. Review Comments to the Author

Reviewer #1: (No Response)

Reviewer #2: Authors have revised the manuscript carefully and incorporated all the comments and suggestions accordingly. Thus I recommend that accept this manuscript for publication.

7. PLOS authors have the option to publish the peer review history of their article (what does this mean?). If published, this will include your full peer review and any attached files.

Reviewer #1: **Yes: **Tusawar Iftikhar Ahmad

Reviewer #2: **Yes: **QASIR ABBAS

---

## [Editor Report · Acceptance letter]

9 Aug 2024

PONE-D-24-02693R1 

PLOS ONE

Dear Dr. Msuha, 

I'm pleased to inform you that your manuscript has been deemed suitable for publication in PLOS ONE. Congratulations! Your manuscript is now being handed over to our production team.

Kind regards, 

on behalf of

Dr. Muhammad Khalid Bashir 

Academic Editor

PLOS ONE